# The Magnitude of the Frequency Jitter of Acoustic Waves Generated by Wind Instruments Is of Relevance for the Live Performance of Music

**Alexander M. Rehm** 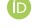

Institut für Strukturanalyse/Frankfurt, Holzhausenstraße 48, 60322 Frankfurt, Germany;
alexander_rehm@gmx.de or arehm@ifsa-gmbh.de

**Abstract:** It is shown that a gold-plated device mounted on a tenor saxophone, forming a small bridge between the mouthpiece and the S-bow, can change two characteristics of the radiated sound: (1) the radiated acoustic energy of the harmonics with emission maxima around 1500–3000 Hz, which is slightly reduced for tones played in the lower register of the saxophone; (2) the frequency jitter of all tones in the regular and upper register of the saxophone show a two-fold increase. Through simulated phase-shifted superimpositions of the recorded waves, it is shown that the cancellation of acoustic energy due to antiphase superimposition is significantly reduced in recordings with the bridge. Simulations with artificially generated acoustic waves confirm that acoustic waves with a certain systematic jitter show less cancelling of the acoustic energy under a phase-shifted superimposition, compared to acoustic waves with no frequency jitter; thus, being beneficial for live performances in small halls with minimal acoustic optimization. The data further indicate that the occasionally hearable "rumble" of a wind instrument orchestra with instruments showing slight differences in the frequency of the harmonics might be reduced (or avoided), if the radiated acoustic waves have a systematic jitter of a certain magnitude.

**Keywords:** wind instruments; frequency jitter; power spectra; acoustic radiation; harmonics; superimposition



## 1. Introduction

Several professional musicians playing wind instruments (i.e., brass and woodwind), and musical conductors of wind instrument orchestras, have started to regularly use a metal device mounted on the instrument in such a way that it forms a bridge between the mouthpiece and the S-bow (or the body) of the wind instrument (see Figure 1 of gold-plated sound bridge mounted on a tenor saxophone).

According to the manufacturer (lefreQue BV/Netherlands), the general concept of the sound bridge is to facilitate the transfer of vibrational energy from one part of the instrument to the next connected part. It is assumed that, at the first loose connection between the mouthpiece and the body of the wind instrument (e.g., the S-bow of a saxophone or lead-pipe of a trumpet), a loss in the transfer of vibrational energy occurs, which can be reduced through the use of a metal bridge, which is designed to transfer vibrational energy (see schematic drawing in Figure 2).

Although a recent study did not detect significant perceptual differences with or without such a bridge mounted on a trumpet [1], the arguments of professional musicians and musical conductors in favor of the use of such metal bridges are in contrast with these findings. Two typical statements are as follows:

(1). "Playing solo I have more projection in my sound and especially in small to medium size halls people can hear me more clear, even if they are in my back" and;
(2). "With the bridge the wind instruments are playing more in tune; the typical rumble especially of non-professionals playing in an orchestra can be avoided".

A comparison of the spectral analysis of tones played on a piccolo flute with and without a sound bridge showed a shift of the harmonics towards higher frequencies due to the use a mounted bridge [2]. As no further data were presented, this finding does not help to explain the impressions of musicians and conductors described above. As professional musicians and experienced musical conductors hear a difference, obviously caused by the metal bridge mounted on the wind instruments, it can be concluded that the emitted acoustic waves must differ after mounting the bridge. Therefore, it is worth to make attempts to:

(a) Determine acoustic parameters, which correlate with this audible difference and;
(b) Uncover the physical process behind these observed phenomena.

Tenor saxophone with mounted Gold-plated bridge

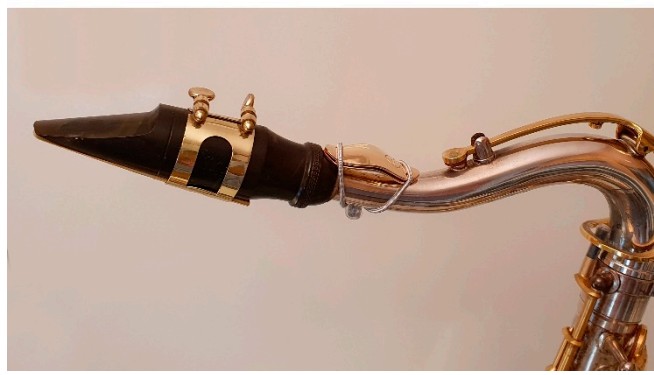

**Figure 1.** Picture of the upper part of a tenor saxophone with a "Gold-plated sound bridge" mounted to the connection-point of mouthpiece and S-bow according to the instructions of the manufacturer of the sound bridge. Source of picture: lefreQue BV/Netherlands.

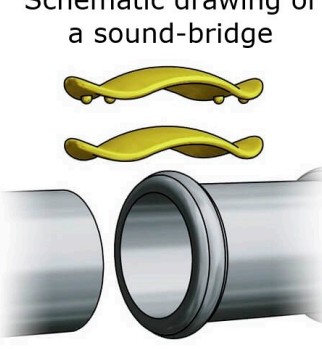

**Figure 2.** Schematic drawing of the design of a "sound bridge" consisting of two equally shaped metal parts (upper part of graphic). Mounted to a wind instrument as shown in Figure 1 it generates a "bridge-like" connection of the mouthpiece and the next part of the instrument (e.g., the S-bow of a saxophone or lead-pipe of a trumpet).Source of drawing: lefreQue BV/Netherlands.

Following the fundamentals of acoustics, the mounting of such a bridge must influence and modify the radiated acoustic wave of a played wind instrument, in order to generate a difference in sound, which can be heard. A change in intonation as a consequence of mounting the metal bridge can be excluded from causing the audible effect, as players tune their instrument with the bridge mounted. Thus, the effect must result from changes in other characteristics of the radiated acoustic wave. It has been reported that professional saxophone players can adjust their vocal tract to achieve and optimize the output of the generated acoustic radiation [3–6] and, in general, have a high capability to influence their sound [7–9]. Other parameters, such as the expression of certain formants and noise, as part of the sound generated by the player, might be affected by the mounting of a bridge

on the instrument. Short time fluctuations of the emitted acoustic energy (shimmer) or of the audible and measurable frequency (jitter) may undergo changes due to the use of a mounted bridge. As it has been shown that these parameters are relevant for the emitted sound and are controlled, to some extent, by professional players, it might be the case that a mounted bridge either affects these parameters directly or influences the interaction between the player and the instrument, which may result in changes in the above-mentioned parameters. In this study, the jitter of the played tone is investigated in detail, as significant changes in this parameter could be detected with the use of a mounted bridge on a tenor saxophone. The aim is to identify which variations of the acoustic waves are induced by the mounted bridge, and whether these variations might be useful to propose an explanation for the audible effects described by professional musicians and conductors. Based on recordings with experienced tenor saxophone players it is demonstrated in this study that a sound bridge mounted to the saxophone (see Figure 1) results in a significant increase of the frequency-jitter of the radiated acoustic waves. The relevance of this effect for the live performance of music where complex superimposition of acoustic waves radiated by the same or different instruments may occur is further evaluated and discussed.

## 2. Materials and Methods

The playing, recording, and analysis of acoustic waves, and the determination of acoustic parameters such as power spectra (frequency-dependent intensity spectra), player noise, formant expression, intensity shimmer, and frequency jitter of the recorded acoustic signals were performed under conditions and with equipment as described previously [7–9]. The recordings of the acoustic waves were carried out using the following equipment: Rode NT5 microphone (Rode, Silverwater, Australia), Behringer XENYX 1204 USB Mixer (Behringer, Willich, Germany), Mixcraft 8 Pro Studio recording software (Acoustica, Oakhurst, CA, USA), and Praat analysis software [10]. The equipment was set up to record pure signals without any effects; this set-up was unaltered during all recordings. The positioning of the saxophone vs. the microphone was defined and remained unchanged during all recordings. Segments 1 s in length of these recordings (wav files with a sample rate of 44 kHz) during stable tone generation were used for analysis with Praat. Power spectra were generated using the fast Fourier transformation (FFT) function [11,12] of Praat, and displayed as frequency (unit = Hz)-dependent intensity (unit = dB) spectra (power spectra). Player noise was calculated from the FFT spectra by measuring the dB minima located between the dB maxima, which can be attributed to the harmonics of the recorded tone. The expression of formants (formant spectra) of a recording was performed by subtracting a calculated decay curve from the power spectrum [9]. The shimmer of the intensity (variation in dB) and the jitter of the frequency (variation in Hz) were measured using the respective functions in the Praat software. The basic recordings with the tenor saxophones were performed in a room equipped with damping material, in order to minimize the effects due to reflection or superimposition of generated acoustic waves. The figures were generated using the respective functions in the Excel and Praat software. The logarithmic functions visible in the presented power spectra and the linear regression functions visible in the presented figures were calculated through mathematical trend analysis, using functions in the Excel software. The resulting logarithmic functions in the power spectra were considered, in order to describe the principal distribution of the overall radiated acoustic energy in the harmonics of the tone played. The following sound bridges from the company lefreQue BV (TJ Hoogland, The Netherlands) were used: Solid-Silver 41 mm (article number: 164125); and gold-plated 41 mm (article number: 164130).

The silver and gold-plated bridges were mounted on the mouthpiece and the S-bow of the saxophone, according to the guidelines of the manufacturer (lefreQue BV). Repeated recordings with and without the mounted bridges were performed by one professional saxophonist (playing a Selmer balanced Action and Otto Link metal mouthpiece) and two experienced but non-professional saxophone players (King Super 20 Silver Sonic and

Otto Link hard rubber MPC; RS Berkley Virtuoso and Otto Link hard rubber MPC). The saxophonists were asked to play a defined number of notes as stably as possible for 2 s, with or without a mounted bridge, by keeping their regular embouchure and blowing pressure as constant as possible. For the comparison of recordings of one player with or without a mounted sound bridge, it was assured that the total radiated energy (calculated by Praat) of the analyzed segments (with a length of one second) differed by a maximum of 10%. This is of high importance, as significant differences in the level of the radiated energy (which may be caused by a variation of blowing pressure) result in variations in the power spectra which cannot be attributed to the mounted bridge.

Processing of the wav files and shifting the phase and/or superimposing the acoustic waves was performed with the Praat [10] and WavePad Master's Edition software (NCH, Greenwood Village, CO, USA).

Superimpositions of phase-shifted acoustic wave signals were simulated by simple mathematical addition of the digital signals of the respective wav files and adjustment of the intensity (amplitude) of the superimposed waves, accordingly. Files were processed to generate a certain phase shift.

A simplified room and radiation model was developed using previously published data [13,14].

Artificial acoustic sine waves and artificial sounds (multiple sine waves consisting of different frequencies) were generated using a specific function in Praat [10].

## 3. Results

Under the experimental setup described in the Material and Methods section, the mounting of a gold-plated bridge generated significant changes in two of the investigated parameters: (1) the distribution of the radiated energy among the harmonics (including the base frequency) displayed in the power spectra and (2) the frequency jitter of the radiated acoustic waves. For the other investigated parameters (formant expression, player noise, and shimmer of the intensity), no significant and reproducible effects could be detected with use of the gold-plated bridge. The effects of the silver bridge on all investigated parameters were, in general, small and did not show clear trends; therefore, they were not considered to be significant.

### 3.1. Power Spectra (Frequency Dependent Distribution of Radiated Acoustic Energy)

The power spectra of the played tones A (196 Hz) and B (220 Hz) on the Tenor Sax showed minor but significant changes in the range of 1000–3000 Hz with use of the gold-plated bridge, whereas no significant changes could be detected when the tones were played an octave higher (392 Hz and 440 Hz). For the played tone A, Figure 3 demonstrates that the overall distribution of the radiated energy among the harmonics in the range of 196–9000 Hz did not differ significantly after mounting the bridge. This can be concluded from the similar factors of the respective Log-functions (−16.85 vs. −17.03).

In the range of 196–3200 Hz, a significant and reproducible difference in the distribution of radiated energy caused by the bridge could be detected (Figure 4). The differences in the factors of the respective Log-functions (−9.4 with bridge vs. −7.74 without) indicated that mounting the bridge resulted in a relative reduction of radiated energy in the range of 1000–3200 Hz, compared to radiated energy <1000 Hz. In the range of 3000–5000 Hz, slight variations of the power spectra were also detected among all the players; however, there was no clear trend and, thus, these variations were considered minor and were not attributed to the mounting of a sound bridge.

The observed difference in the distribution of radiated energy was a small, but significant effect of the mounted gold-plated bridge, as several recordings that varied by less than 10% in the amount of total radiated acoustic energy confirmed this phenomenon. As human beings have a high sensitivity to acoustic signals in the range of 1500–3000 Hz, it can be assumed that this change in the distribution of radiated energy caused by the bridge

is an audible effect, especially if the listeners are professional musicians or experienced musical conductors trained to recognize small sound differences.

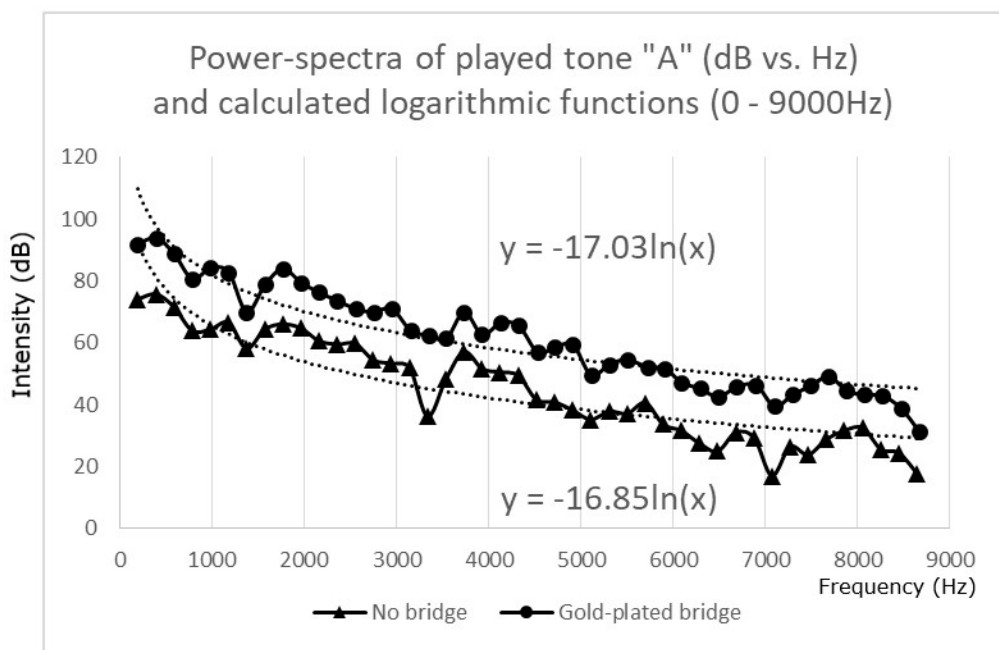

**Figure 3.** Intensity maxima (dB) of the power spectra of played tone A (base frequency, 196 Hz) with a mounted bridge or without bridge and respective calculated logarithmic functions of the displayed data in the range of 196–9000 Hz (see Material and Methods). Intensity values with the mounted bridge are shifted by +20 dB on the *y*-axis, for better visibility of the data. As such, the upper displayed function represents the data with a mounted bridge.

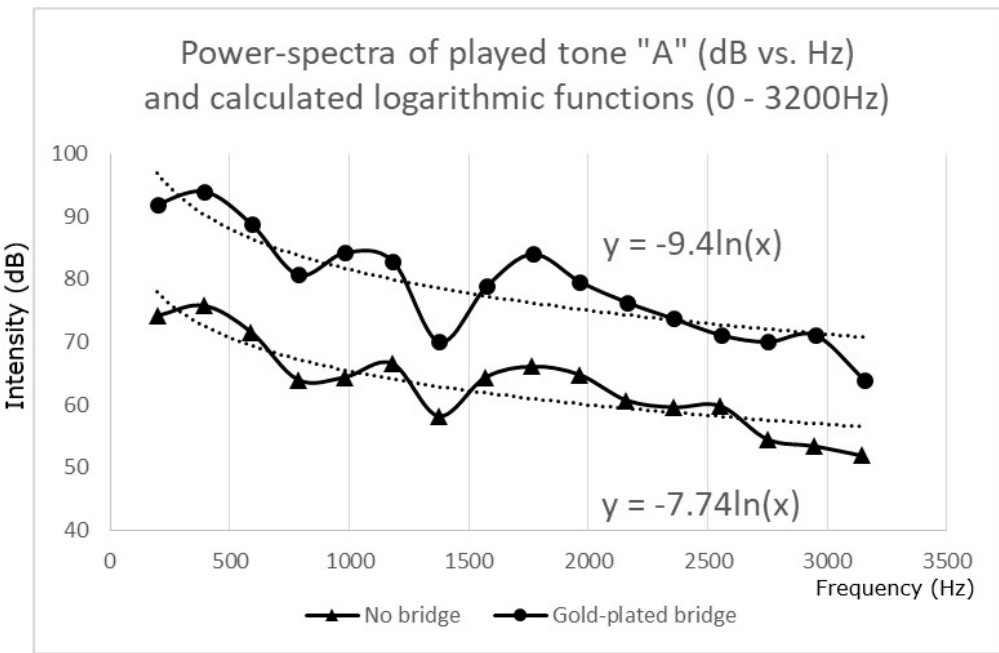

**Figure 4.** Same data as in Figure 1 with respective calculated logarithmic functions in the range of 196-3200 Hz (see Material and Methods). Intensity values with mounted bridge are shifted by +20 dB on the *y*-axis, for better visibility of the data. As such, the upper displayed function represents the data with a mounted bridge.

### 3.2. Frequency Jitter (Distortion of the Basic Frequency of the Acoustic Waves)

The frequency jitter of the acoustic waves generated by playing the saxophone showed significant changes as a result of mounting the gold-plated bridge on the saxophone. Table 1 summarizes the values for the standard deviation of the basic frequency for the tones A, B, C#, Oct-A, Oct-B, and Oct-C#, played stably for one second on the tenor saxophone. As the value of the standard deviation provides an indication of the range of the frequency jitter of the radiated acoustic wave, these data demonstrate that the frequency jitter of all tones played (spanning a range for the base frequency of 196–498 Hz) was approximately doubled, as a result of mounting the gold-plated bridge.

**Table 1.** Standard deviation of the frequency jitter of tones played on the saxophone with or without a mounted gold-plated bridge.

| | Standard Deviation of Jitter (Hz) | | |
|---|---|---|---|
| Tone | Gold bridge | No bridge | Ratio |
| "A" | 0.34 | 0.17 | 2.0 |
| "B" | 0.32 | 0.12 | 2.7 |
| "C#" | 0.42 | 0.26 | 1.6 |
| "Oct-A" | 0.99 | 0.46 | 2.2 |
| "Oct-B" | 0.99 | 0.55 | 1.8 |
| "Oct-C#" | 0.71 | 0.49 | 1.4 |
| | | Average: | 1.9 |

The differences (in Hz) between the highest and lowest frequency for a single turnover within the period of one second of stable tone generation (as calculated by Praat) confirmed the standard deviation data and underlined the finding that the gold-plated bridge increased the jitter of the basic frequency of a played tone significantly. Figures 5 and 6 further show that the harmonics of the played tones underwent the same changes in the jitter as the basic frequencies, as a result of mounting a gold-plated bridge. The displayed data further demonstrated that, under the experimental setup, the effect of the silver bridge on the frequency jitter of a played tone was either not detectable or very small.

### 3.3. Effects of Superimposition of Acoustic Waves of Played Tones

Superimposition of acoustic sine waves of the same frequency can increase the signal (if both waves are in phase), or may result in total cancellation of the signal (antiphase superimposition). The systematic frequency jitter of superimposed waves should have an effect on the resulting acoustic signal. The effect of superimposition of the acoustic wave generated by playing a B on the saxophone with the same acoustic wave, but shifted by a time span of 2500 Hz (0.0004 s), on the power spectrum is displayed in Figure 7. As expected, a significant reduction of the acoustic signal with a maximum effect at approximately 1250 Hz can be demonstrated.

Superimposition of an acoustic wave with the identical waveform, shifted by 0.0004 s in phase, excluded any effect of systematic frequency jitter, as each single turnover was compared to itself; as such, time distortion of the basic cycle, as a possible mechanism for a frequency jitter, did not influence the result of the superimposition. Superimposing phase-shifted acoustic signals of the same frequency and form, but with different jitter, should result in lower reduction of the dB signal in the range of 1250 Hz. As demonstrated in Figure 8, cancellation of signals in the range of 1250 Hz after superimposition of phase-shifted acoustic waves was reduced when two waves of different recordings of B without a mounted bridge were superimposed. Only a minor cancellation was detectable when the superimposition was carried out using recordings of the same tone with a mounted gold-plated bridge. This indicates that the increase in frequency jitter through a mounted bridge may be of importance under conditions where multiple reflections (and, therefore, superimpositions) of the generated acoustic waves take place, as is the case in small-to medium-sized rooms with minimal acoustic optimization.

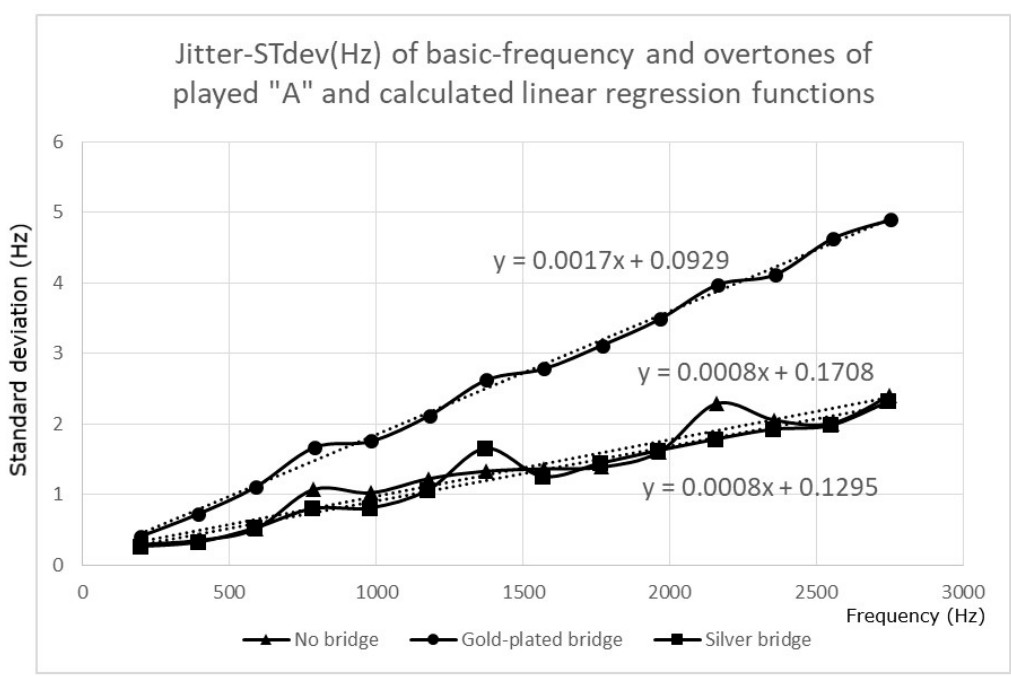

**Figure 5.** Standard deviation of the frequency Jitter of the basic frequency and the related overtones (harmonics) of a played tone-A (196 Hz)on the saxophone with and without metal bridges mounted. The dashed lines represent the linear regression of the displayed data, with their respective linear function (see Material and Methods). The upper displayed function represents the linear regression of the data of the gold-plated bridge; the lowest function represents the regression of data of the silver bridge.

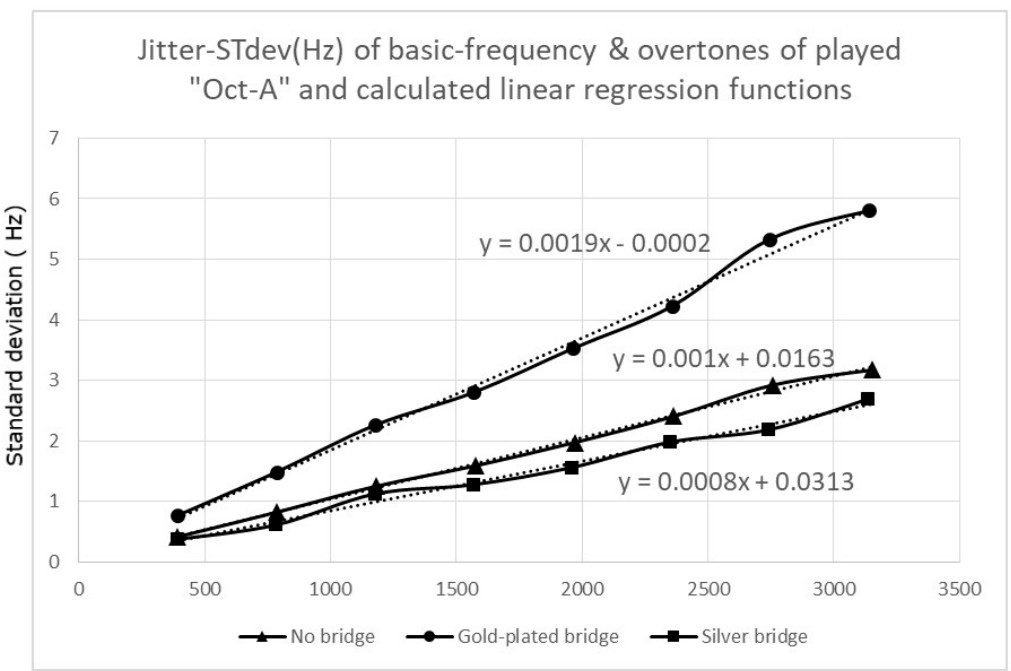

**Figure 6.** Standard deviation of the frequency jitter of the basic frequency and the related overtones (harmonics) of a played tone-Oct-A (392 Hz) on the saxophone with and without metal bridges mounted. The dashed lines represent the linear regression of the displayed data, with their respective linear function (see Material and Methods). The upper displayed function represents the linear regression of the data for the gold-plated bridge, while the lowest function represents the regression of data for the silver bridge.

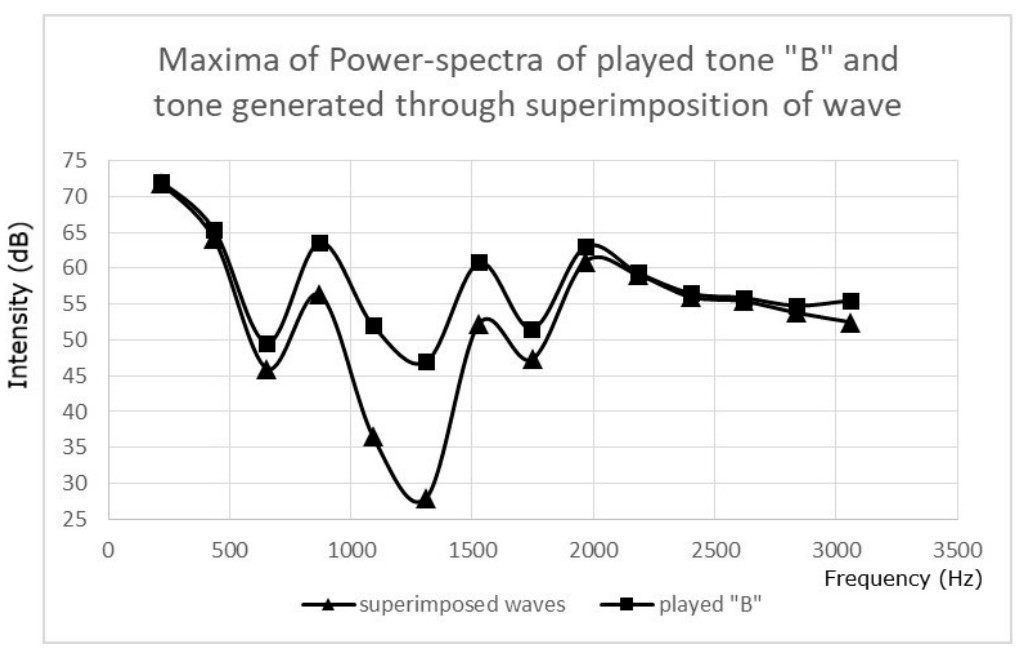

**Figure 7.** Power spectra of tone B (220 Hz) played on the saxophone and of a wave generated through superimposition of this original wave with itself, but shifted in phase by 2500 Hz (0.0004 s). The intensity of the generated wave was reduced to 50%, in order to compare the power spectra.

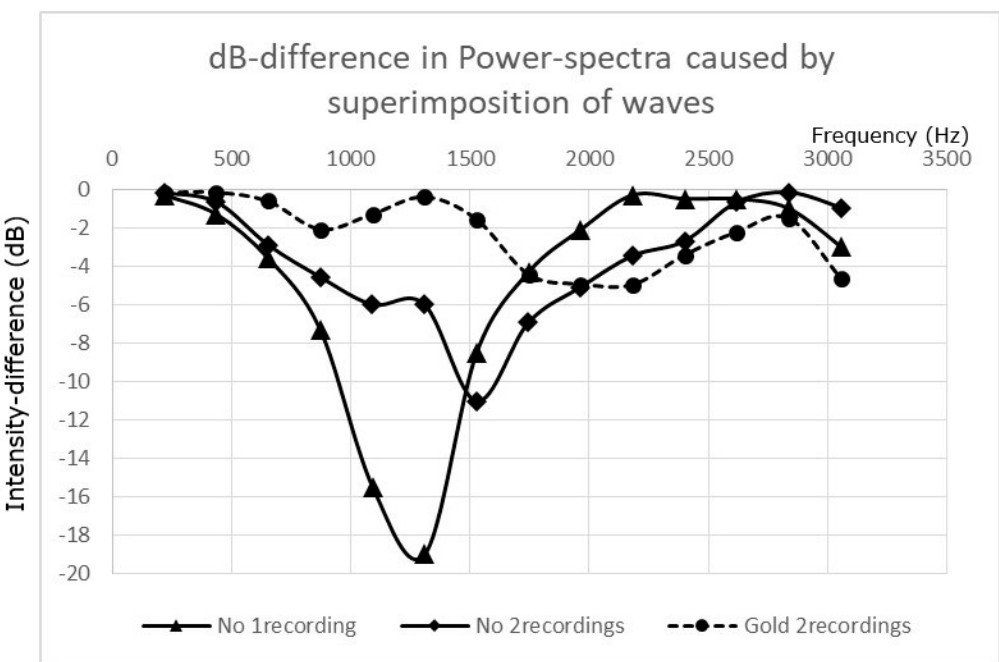

**Figure 8.** Differences in the power spectra of waves generated through superimpositions of original waves with waves shifted in phase by 2500 Hz. No. 1 recording: no bridge mounted and superimposition with phase-shifted same recording of played tone B; No. 2 recordings: no bridge mounted and superimposition with a second phase-shifted recording of played tone B; Gold 2 recordings: gold-plated bridge mounted and superimposition with a second phase-shifted recording of played tone (B) with the bridge mounted.

*3.4. Artificial Creation of Acoustic Waves with Frequency Jitter and Effects of Superimposition of These Waves*

Although the presented data deliver strong arguments that (a) mounting the gold-plated bridge on the saxophone increased the frequency jitter of the played tone sig-

nificantly and (b) superimposition of phase-shifted acoustic waves generated different effects when the jitter of the superimposed signals differed, these findings did not simply explain the "heard perceptions" of professional musicians or musical conductors (see the Introduction section). With the Praat software, acoustic waves can be created using simple mathematical formulae; for example, by using the mathematical operation $0.1 \times \sin(2 \times pi \times 220 \times x)$, a sine wave with frequency of 220 Hz and amplitude of 0.2 can be created. Measuring the frequency jitter of this signal provided a value for the Jitter range of 0.024 Hz with a standard deviation of 0.009 Hz. In theory, the jitter of this artificially generated wave should be 0; therefore, the jitter values calculated by the software can be understood as noise or measurement error. With the following formula, a 220 Hz wave with a certain jitter can be generated in Praat: $0.1 \times \sin(2 \times pi \times (220 + 1 \times \mathrm{sinc}(2 \times pi \times 1 \times x)) \times x)$. This wave has a jitter range of 2.001 Hz and a standard deviation of 0.689 Hz. Modifying the formula allows for the generation of waves at any frequency with various magnitudes of Jitter. It is worth noting that, with this method, a systematic jitter is generated, which, itself, follows a sine function.

For the superimposition simulation, an acoustic wave consisting of signals at 220 Hz, 440 Hz, 660 Hz, and 880 Hz with artificial jitter was created. The jitter range of the 220 Hz signal was set to 2 Hz; while those of the 440, 660, and 880 Hz signals were set to 4, 6, and 8 Hz, respectively. The effect of the superimposition simulation with waves shifted in phase by 1000 Hz on the power spectra of the resulting acoustic waves is shown in Figure 9. It is obvious that the superimposition of such a wave by the same wave with the same artificial jitter, but shifted in phase by 0.001 s (1000 Hz), resulted in the expected strong reduction of the signal, with a maximum effect in the range of 500 Hz. No signal reduction occurred when the type of artificial jitter in the superimposed waves differed.

**Figure 9.** Power spectra of artificially generated acoustic waves with signals at 220, 440, 660, and 880 Hz, generated using the Praat software. Solid line: Original wave with a frequency jitter of 2 Hz at 220 Hz. Dotted line: Original wave superimposed by itself with a phase shift of 1000 Hz (0.0001 s). Dashed line: Original wave superimposed by a phase-shifted (1000 Hz) wave with identical parameters, but with a different systematic jitter than the original wave. For better visibility, the *x*-axis is shifted by 20 Hz for each spectrum.

The displayed results of the simulation were very similar to the recognized effects of the superimposition experiments with the acoustic waves of the saxophone having the gold-plated bridge vs. that with no bridge mounted (see Section 3.3 above). These findings support the idea that a change in the extent of the frequency jitter may reduce the regular cancellation of the amplitude of the radiated (and heard) sound, caused by a superimposition of phase-shifted waves due to substantial reflection phenomena in closed rooms.

The observation of experienced musical conductors—that a wind orchestra of laymen players often generates a "rumble"—is mainly due to the fact that the instruments are well tuned for the base frequency (e.g., 220 Hz), but may differ in the frequencies of related overtones. Such an effect can be easily simulated through the superimposition of acoustic waves, which are identical in base frequency, but slightly differ in the frequencies of the next three overtones. Power spectra of superimpositions of acoustic waves having signals at a) 220, 441, 662, and 883 Hz and b) 220, 439, 658, and 877 Hz, with and without artificial frequency jitter, are shown in Figure 10. As expected, the superimposition of two different acoustic signals having no artificial frequency jitter resulted in a power spectrum with clear double signals at 662–658 Hz and 883–877 Hz, mimicking the audible rumble of the sound. Furthermore, when the two superimposed signals had an artificial frequency jitter of the same type and magnitude, similar double signals were observed in the power spectrum. However, when the artificial frequency Jitter of the superimposed signals systematically differed, the resulting acoustic wave did not show double peaks but, instead, distinct peaks in the power-spectrum at 440, 660, and 880 Hz, such that no rumble will be generated—a phenomenon that is comparable to the experience of musical conductors after mounting bridges on the wind instruments of the orchestra (see the Introduction section).

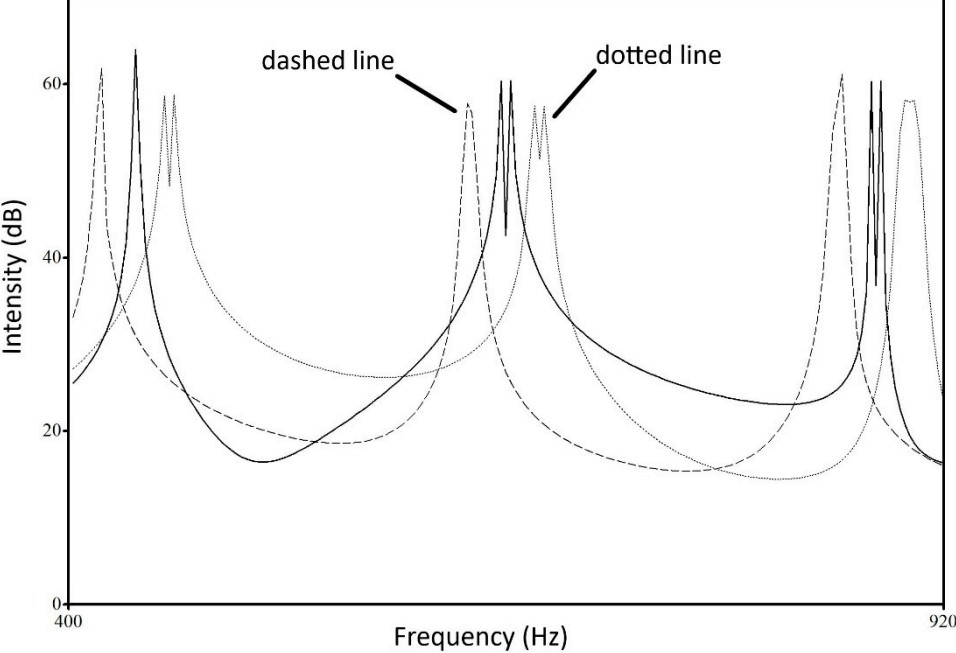

**Figure 10.** Power spectra (generated and displayed using Praat) of acoustic waves generated by superimposition of two artificially generated waves having the same basic frequency (220 Hz), but with different frequencies for the next three overtones (441, 662, and 883 Hz and 439, 658, and 878 Hz). The two artificial waves used for the superimposition had either no systematic frequency jitter (solid line), an identical systematic frequency jitter (dotted line), or a different systematic frequency jitter (dashed line). For better visibility, the *x*-axis of the spectra with frequency jitter is shifted by 20 Hz.

*3.5. Modeling the Reception of Intensity and Sound Characteristic of an Acoustic Signal Radiated by a Saxophone in a Medium-Sized Room by a Listener*

The following experimental set-up and observation mimicked the experience of professional wind instrument players playing in a medium-sized room with limited acoustic optimization (see Introduction section):

In a medium-sized room (dimensions: 14 m $\times$ 9 m $\times$ 2.7 m) with regular walls (no acoustic optimization), a listener was placed in the middle of the first half of the room, opposite to a saxophone player placed in the middle of the second half of the room. First, the player was face-to-face to the listener; then, they turned their back to the listener (face-to-back) while the same tone sequence was played with the same intensity. This procedure is repeated several times, and the player randomly mounted the bridge to their instrument or removed the bridge again. The listener had no information and could not see whether the bridge was mounted (as the part of the saxophone where the bridge is mounted was covered during the entire experiment), but knew whether the player position was face-to-face or face-to-back. It was confirmed by all listeners participating in this experiment that, in certain cases, they recognized a tremendous change in the sound (mainly in intensity) when the player turned from a face-to-face to a face-to-back position; whereas, in other cases, they recognized less changes in sound in the face-to-back position vs. the face-to-face position. The latter impression correlated 100% with a bridge mounted on the saxophone. Although this experimental set-up lacked some precision, and more sophisticated experiments are necessary (which will be executed in the near future) to quantify the audible effect, the reports of musicians playing in medium-sized rooms were principally confirmed (see the Introduction section).

One way to explain this phenomenon is to consider a different magnitude of cancellation of acoustic energy by superimposition of acoustic waves with more or less frequency jitter. Further, the three-dimensional spherical radiation pattern of acoustic energy of a saxophone (including the player) is of importance, as the majority of acoustic energy is emitted to the area in front of the player, while only a minor portion is emitted to the area behind the player [13]. With a simplistic room and radiation model (for a medium size room, using the data published in [12,13]), simulation of the radiated acoustic energy reaching the ears of the listener (thus contributing to their impression of the change in sound) can be carried out. In this model, the variables are: (1) the absorption and (2) reflection of acoustic energy by the walls and by the body of the player and (3) the cancellation of acoustic energy due to superimposition of reflected (and, therefore, phase-shifted) acoustic waves with or without frequency jitter (simulating a mounted bridge vs. no bridge). For the face-to-face orientation of the musician and listener, the model delivered very similar values for the portion of acoustic energy reaching the ears of the listener (61-66% with frequency jitter assumed and 58–59% without frequency jitter). In the face-to-back orientation of the listener and player, the model yielded strong differences: 34–36% of acoustic energy with frequency jitter and 19% without frequency jitter reached the ears of the listener. This indicates that, due to the frequency jitter (simulating a mounted bridge), close to 60% of the intensity of an acoustic signal reaching the listener would remain, even if the player turns their back towards the listener; whereas, without frequency jitter (simulating no bridge mounted), only one-third of the signal intensity will reach the ears of the listener in the face-to-back position. Such a discrepancy in acoustic energy reaching the ears of the listener may significantly influence their perception of sound. Although the model is a simplification of reality, it delivers results that can be used to explain the reported impressions of listeners.

## 4. Discussion

It was demonstrated that mounting a gold-plated bridge on a tenor saxophone can influence the distribution of radiated acoustic energy for the played tones A (196 Hz) and B (220 Hz) in the frequency range of 1000–3200 Hz. Although the gold-plated bridge generated a slight, but significant variation of the distribution of the radiated energy in the range up to 3200 Hz (see Figure 4), it may be enough to be perceived by an experienced

listener. A reduction of radiated acoustic energy in the range of the "Singer formant" could be of general benefit when playing in a wind instrument orchestra, as the orchestra may sound more homogenous without excessively pronounced overtones in the frequency range of 1000–3000 Hz [15]. Although no effects could be detected with the silver bridge, it cannot be excluded that, in a different setup with other instruments, significant effects of a mounted silver bridge on the distribution of the radiated acoustic energy may occur. Further research is needed to evaluate whether combinations of different materials of the bridge and different materials of the instruments generate significant changes in the distribution of the radiated acoustic energy on the harmonics.

Whereas the effect of the gold-plated bridge on the distribution of the radiated acoustic energy was small and limited to tones in the lower register of the saxophone, an approximate two-fold increase in frequency jitter due to the mounted gold-plated bridge was observed over the entire regular tone range of the tenor saxophone (see Table 1), with the same linear function among the respective harmonics (see Figures 5 and 6). This serves as a strong indicator that the mounted gold-plated bridge influences the radiated acoustic signal of a tenor saxophone by increasing the frequency jitter of the radiated acoustic wave systematically. Although it has been reported that professional saxophonist have a high capability to influence the sound parameters, especially the frequency jitter [7] and being able to tune their vocal tract during play [3–6], it is very unlikely that, in this experimental set-up, the players generated the measured effect with a mounted gold-plated bridge, for two main reasons: (1) the effect could also be detected with non-professional players, who have less capability to influence the sound and to tune their vocal tract and (2) no significant effect was detected when using a mounted silver-plated bridge. It is worth noting that all three players in this study reported that they had the impression that the effect of the gold-plated bridge on the sound was clearly audible, whereas the effect of the silver bridge was either not audible (one player) or small, compared to the gold-plated bridge (two players). Therefore, a certain bias of the players cannot be excluded; however, the consistency and high reproducibility of the data serve as strong indicators that the observed effect on the frequency jitter was caused by the gold-plated bridge.

Although the underlying mechanism for this increase in frequency-jitter due to a mounted metal bridge is not yet understood, it is worth considering either (1) a transfer of vibrational energy from the mouthpiece through the bridge to the saxophone (or vice versa) or (2) the damping of vibrational energy within the mouthpiece and/or S-bow, due to the mounted bridge, causing the observed effect.

As the conical form of the saxophone is responsible for the general distortion of the standing wave [16], the observed increase in the frequency jitter could be interpreted as an additional, fluctuating distortion of the standing wave within the saxophone. Several studies have investigated the oscillation of the body and the bell of a trumpet during play [17–19]. The proposed mechanism is a coupling of axial bell vibrations to the internal air column, which influences the radiated signal. If it is assumed that the mounted bridge may facilitate the transfer of vibration energy from the mouthpiece to the body (which is regularly hindered due to the damping effect of the cork between mouthpiece and S-bow or body), a change in the oscillation pattern of the saxophone body and bell might be the result, which may generate an increase in frequency jitter through fluctuating distortion of the standing wave. A damping effect of the mounted bridge causing the observed increase in frequency jitter seems to be very unlikely, for the following reasons: (a) the mass of the metal sound bridge is very small compared to the mass of the tenor saxophone; (b) the sound bridge is loosely fixed to the instrument; and (c) the bridge is mounted far away from the bell, which has been shown to be a prerequisite for a measurable and audible damping effect [19]. Further investigations are needed, in order to understand the underlying mechanism of the observed phenomenon, as it cannot be excluded that slight changes of the vibrations in or close to the mouthpiece may lead to a sophisticated feedback reaction of the player, which may cause or accelerate the observed effect. Experiments with

artificial blowing machines may help to investigate the role of the player in creating the measurable and audible effects.

Simulations have shown that superimpositions of acoustic waves with identical base frequency but with differing frequencies of the related harmonics will create an audible rumble, unless the two different acoustic signals have a substantial and different systematic frequency jitter (see Figure 10). Such a situation could also be expected for the different wind instruments in an orchestra. The results of these simulations may be used to explain the reported impressions of musical conductors of wind instrument orchestras, with respect to the effects of mounted bridges (see the Introduction section).

Considering the data presented on the superimposition of real acoustic signals from a tenor saxophone with a mounted bridge and no bridge (Figure 8), and the artificially created acoustic signals with and without systematic jitter (Figures 9 and 10), it can be concluded that, under conditions where the radiated acoustic signal will undergo various reflections, with resulting superimpositions of radiated and reflected signals (as can be expected in rooms with minimal acoustic optimization; e.g., regular walls), an increased frequency jitter of the radiated signal reduces the cancellation processes of acoustic energy caused by antiphase superimpositions. This effect may explain the differing perceptions of listeners in the face-to-face vs. face-to-back orientations, as the result of a mounted bridge. The output of a simplistic room and radiation model provided further evidence that an increase in frequency jitter of the radiated acoustic waves might be responsible for the audible phenomenon.

Although strong arguments were presented that the magnitude of frequency jitter of an acoustic wave generated by a wind instrument may have significant importance for live performances (especially in medium-sized rooms with minimal acoustic optimization) of solo artists or bands/orchestras, further experiments and more sophisticated simulations and modelling are needed, in order to obtain a better understanding of the processes underlying the observed phenomena.

**Funding:** This research received no external funding.

**Institutional Review Board Statement:** Not applicable.

**Informed Consent Statement:** Not applicable.

**Data Availability Statement:** The data presented in this study (original recordings, raw data, and processed and analyzed data) are available on request from the corresponding author. The data are not publicly available, due to their complex and diverse structure and formats.

**Conflicts of Interest:** The author declares that there are no conflicts of interest.

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
