# Peer review of "The Magnitude of the Frequency Jitter of Acoustic Waves Generated by Wind Instruments Is of Relevance for the Live Performance of Music"

_acoustics, doi:10.3390/acoustics3020027_

Round 1

Reviewer 1 Report

Good article and interesting subject, especially as it deals with feedback phenomena on musicians/users - notoriously complicated... 

There is a lot of interesting information in this paper, but it must be presented more clearly and with more rigour, especially as it deals with a mix of subjective and objective descriptions. 

(1) First main comment: the device is not sufficiently described! Reviewers proposes to add photo or possibly several photos, as well as a full description of what the device is and what it is doing physically. 

Description is currently limited to the word "bridge" (no information whether it is metal, wood, dimensions, etc.), and the words "golden" and "silver". Is it colour only, is it a material change, etc. 

And what is the explanation (whether physical or subjective) why there is an effect with the gold bridge when there is none with the silver bridge? Is it physically different, or is it different user feedback and adaptation??? (Both are possible and both are acceptable, but description needs to be sound). 

(2) minor comment: should read cancellation (or canceling) rather than deletion

(3) other minor comment: definition, better description and reference given for jitter 

(4) major comment: bridge is between mouthpiece and main body of saxophone - so all this happens very close to the mouth and the ears of the player. Main effects are reported on player(s) and conductor. For player things is with the good explanation that feedback to player will influence playing - those effects could be through vibration (transmission from mouthpiece to saxophone body creating better damping of mouthpiece, or transmission from saxophone body to mouthpiece creating stronger vibration in mouthpiece). A conductor is close to the instruments as well. 

So all effects are close-field, with the exception of the experiment with the saxophone player rotating 180° in a room; this is a very different experiment and should be more clearly separated in the text. 

(5) 180° rotation study needs more precise explanation: apparently listeners could see rotation but not bridge - is this true??? Test does not qualify for fully blind test and clearly not as double blind test. Perhaps should be repeated fully double blind or at least as objective as possible, here control of experiment seems too loose to have full confidence in objectivity of the finding. Subjective effects are acceptable...!!! But rigour in experiment is required. 

(6) In figure 1 there is a strong difference in frequency range 3000 to 4000Hz that is never talked about in the text, why?

(7) Page 5, line 134: "measurable Jitter range". 

Is this correct - and why discuss "measurable differences" rather than "audible differences" - which is significant and which do you want to describe?

(8) In general, it would ben interesting to make another experiment to check whether difference from bridge is "objective" (will there be differences with a machine blowing a saxophone) or "player feedback" (which exists as the text explains). Then comes the question whether player feedback is "objective" (through vibration) or actually the equivalent of a placebo effect (which still creates measurable results and changes and which can still make the sound significantly better...). 

(9) page 11 (see discussion above as well): 

This can be due to coupling and changes in acoustic and vibration pattern in the mouthpiece. But could be psychological player changes as well. Would be interesting to try to create some experiments that try to elucidate and get further information. 

Author Response

Dear Reviewer,

I have added my anwers to you comments in the word-file attached. The revised manuscript will be uploaded immediately.

Thanks for you constructive critics and

Best regards

Alex Rehm

Reviewer 2 Report

This a good article that I enjoyed reading. I think this manuscript is acceptable in its current format. 

Author Response

Dear Reviewer,

I have been asked by the editor to make a revision of the manuscript so I will submit the revised version immediately.

I have also decided to use the MDPI service for extensive grammar and spell-check for English language but as this will take approx. 10-14 working days I have decided to submit the revised manuscript prior to this check so you can review the changes already.

Thanks again for your support to improve the quality of the paper.

Best regards

Alex Rehm

Reviewer 3 Report

In this paper the authors investigate on parameters like formant-expression, Player-noise and Shimmer & Jitter of the generated acoustic waves in order to develop an understanding on which changes of the acoustic waves are induced by a mounted bridge and whether the “heard effect” could be explained by the observed changes.

Section 1 must be improved. You should introduce the problem in more detail so that the reader is immediately clear about the purpose of your study. Specify better the essential elements of the problem. You should add more information in the introductory part, you should add other works that have also addressed the problem. You must properly introduce your work, specify well what were the goals you set yourself and how you approached the problem. At the end of the section, add an outline of the rest of the paper, in this way the reader will be introduced to the content of the following sections.

Section 2 must be improved. It is not enough to quote your works, however in German, to describe materials and methods used in this work. You must also adequately introduce the parameters that you will then analyze: Power-spectra (frequency dependent intensity spectra), Player- noise, Formant-expression, intensity-Shimmer and frequency-Jitter of the recorded acoustic signals. Otherwise, the reader when he goes to analyze the results will not be able to follow the flow of information. This section is just for this, so briefly describe which materials and which methodologies you used and then add references to your other works to allow the reader to deepen the topic. Add pictures of the equipment you are testing and the equipment you used to detect the signals. The teacher must explain in detail the initial conditions of the problem. How do you know if the musicians have blown the same amount of air? How did you measure this parameter? This is crucial because later I saw you compare the results, but if you are not sure if the same initial conditions were used then it is not possible to make a comparison.

Section 3 must be improved. Here arises the doubt raised earlier. How can you be sure that the two measures refer to the same initial conditions? How much air did they blow into the instrument? How did you manage to secure this condition with a small margin of error? You have to specify it in detail.Try to enrich the captions of the figures, the reader should be able to read the figure without the need to retrieve the information in the paper. Try to summarize the essential parts of the Figure and what you want to explain with it.

Section 4 must be improved. Paragraphs are missing where the possible practical applications of the results of this study are reported. What these results can serve the people, it is necessary to insert possible uses of this study that justify their publication. They also lack the possible future goals of this work. Do the authors plan to continue their research on this topic?

30) change (1) with [1]

32-36) Show this as bullet list

38-40) Show this as bullet list

58) It is not enough to quote your works, however in German, to describe materials and methods used in this work. This section is just for this, so briefly describe which materials and which methodologies you used and then add references to your other works to allow the reader to deepen the topic.

66-72) Add pictures of the equipment you are testing and the equipment you used to detect the signals.

70-72) This is a crucial question of all the work and you need to describe and justify it in detail. How do you know if the musicians have blown the same amount of air? How did you measure this parameter? This is crucial because later I saw you compare the results, but if you are not sure if the same initial conditions were used then it is not possible to make a comparison.
73-84) Describe in detail the methodologies you are using, you have all the space, do not just mention other works. This section is used to describe in detail the methodologies you have used. Then do it.

87-90) Describe in detail the methodologies you are using, you have all the space, do not just mention other works. This section is used to describe in detail the methodologies you have used. Then do it.

95-101) Here arises the doubt raised earlier. How can you be sure that the two measures refer to the same initial conditions? How much air did they blow into the instrument? How did you manage to secure this condition with a small margin of error? You have to specify it in detail.

103) Figure 1: In the caption you must also describe the approximation curve you have drawn and the equation in the graph. You must describe everything there, do not leave this task to the reader.

113) Figure 2: In the caption you must also describe the approximation curve you have drawn and the equation in the graph. You must describe everything there, do not leave this task to the reader.

144) Figure 3: In the caption you must also describe the approximation curve you have drawn and the equation in the graph. You must describe everything there, do not leave this task to the reader.

147) Figure 4: In the caption you must also describe the approximation curve you have drawn and the equation in the graph. You must describe everything there, do not leave this task to the reader.

210) Figure 7: It improves the image quality, also increases the dimensions of the axis labels (text and numbers), making them consistent with the other figures.

240) Figure 8: It improves the image quality, also increases the dimensions of the axis labels (text and numbers), making them consistent with the other figures.

Author Response

Dear Reviewer

I have sent you a separate doc-file with my comments.

Best regards

Alex Rehm

Round 2

Reviewer 1 Report

Thank you for the changes and improvements to the paper, highly appreciated. Paper now ready for publication as it stands. 

Reviewer 3 Report

The author addressed all the comments proposed by the reviewer justifying the points that were not collected with sufficient justification. Now the paper needs format revisions. Starting with the two images that appear in the introductory section and which have not been numbered and which are missing the caption. I have noted below some anomalies that I have found.

31-32) This figure has no caption and therefore will not even be referenced anywhere in the article. it is necessary to add a caption to the Figure and recall it in the paper. Furthermore, the caption must be sufficiently descriptive.

41) This figure has no caption and therefore will not even be referenced anywhere in the article. it is necessary to add a caption to the Figure and recall it in the paper. Furthermore, the caption must be sufficiently descriptive.

54) Add the link to the reference using appropriate notation.

83)  At the end of the Introduction section, add an outline of the rest of the paper, in this way the reader will be introduced to the content of the following sections.

96-97) “Fast Fourier Transform (FFT)” Add references to allow readers to learn more about the topic.

161) Double-check the numbering of the Figures after numbering the first two images contained in the introductory section. Add the axes labels: Frequency (Hz) for x label and SPL (dB) for y label)

178) Double-check the numbering of the Figures after numbering the first two images contained in the introductory section. Add the axes labels: Frequency (Hz) for x label and SPL (dB) for y label)

215) Double-check the numbering of the Figures after numbering the first two images contained in the introductory section. Add the axes labels.

223) Double-check the numbering of the Figures after numbering the first two images contained in the introductory section. Add the axes labels.

241) Double-check the numbering of the Figures after numbering the first two images contained in the introductory section. Add the axes labels.

262) Double-check the numbering of the Figures after numbering the first two images contained in the introductory section. Add the axes labels.

299) Double-check the numbering of the Figures after numbering the first two images contained in the introductory section. Use the same format for all the figures, in this case you have used a bold box.

332) Double-check the numbering of the Figures after numbering the first two images contained in the introductory section. Use the same format for all the figures, in this case you have used a bold box.